# DiT-Serve: An Efficient Serving Engine for Diffusion Transformers

## Abstract

Diffusion Transformers (DiTs) are emerging as a powerful class of generative models for high-fidelity image and video generation, powering highly diverse applications where requests vary in image resolution, video length, and number of denoising steps. Current serving infrastructures largely optimize each request in isolation, missing key opportunities to multiplex GPU compute across requests. Our analysis uncovers two fundamental inefficiencies: spatial underutilization, where GPUs waste compute and memory by padding heterogeneous requests to a common resolution and duration; and temporal underutilization, where batching jobs with varying denoising steps forces GPU cores to idle as shorter requests wait for the longest-running request to finish. We introduce DiT-Serve, an efficient serving engine for image and video models. First, we propose step-level batching, which the scheduler preempts and swaps requests every denoising step, eliminating temporal bubbles. The second innovation is a new attention algorithm, Brick Attention, that binpacks requests of different context lengths onto a set of GPUs, significant reducing padding overhead. Our evaluation over three state-of-the-art models show that DiT-Serve achieves on average 2-3× higher throughput and 3-4× lower latency compared to prior systems.

## 1 Introduction

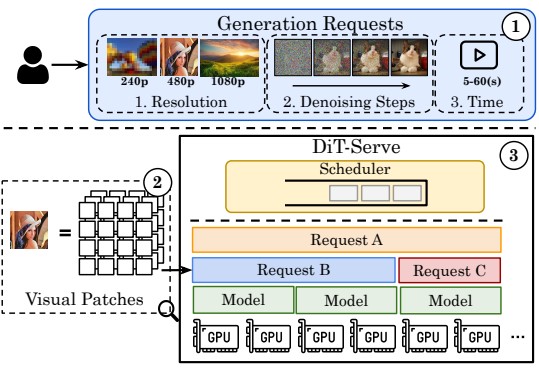

Figure 1: **Serving Diffusion Transformers.** Users submit requests that vary in resolution, time, and the number of denoising steps. Such requests are encoded as long sequences of visual patches/tokens. Finally, DiT-Serve batches user requests and schedules their execution over many GPUs.

Beyond large language models (LLMs) for text domains, diffusion transformers (DiTs) (14) have emerged as powerful, deep generative models for content creation, including high resolution image, video, and 3D model generation (30; 46; 7; 25; 3). DiTs blend 3D Transformer attention—which captures long-range spatial relationships in images and temporal dependencies across video frames—with a diffusion-based denoising process that iteratively turns random noise into increasingly detailed, high-fidelity outputs. As such, diffusion-based models power a wide array of applications—from automated marketing (15) and cinematic editing (35; 6; 28; 31) to social-media content creation (27; 51), personalized entertainment, and high-fidelity restoration and upscaling (13; 2; 50)—transforming how videos are produced, enhanced, and consumed cost-effectively at scale (12; 21). Moreover, such diversity in applications yields highly heterogeneous workloads, with users' requests varying widely in both image resolution, denoising steps, and video duration.

However, serving DiTs at scale remains both technically challenging and cost-prohibitive: the quadratic complexity of self-attention from transformers means that generating a single high-resolution video often takes minutes (20). Innovations like FastVideo (45; 9) and Pyramid Attention (47) mitigate this by caching and reusing attention states across frames, and hardware-aware kernels, such as FlashAttention 3 (36), further accelerate attention computation. Yet, existing work

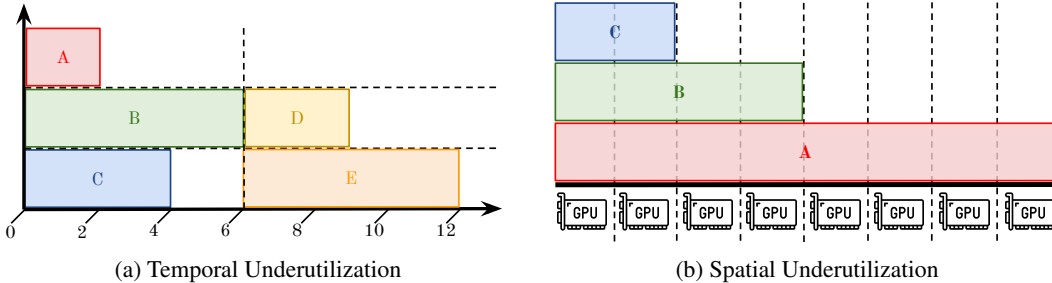

(a) Temporal Underutilization         (b) Spatial Underutilization

Figure 2: **Inefficiences stemming from suboptimal batching.** (a) *Temporal Underutilization*: Batching requests with different decode steps under-utilizes GPUs over time. (b) *Spatial Underutilization*: Due to limitations of existing sequence-parallel algroithm, batching requests of different resolutions and durations leads to excessive padding and hence under-utilization across GPUs.

optimize individual requests in isolation. Critically, few existing diffusion-based serving systems (11) exploit optimization *across* concurrent video requests to better multiplex GPU compute—a strategy that existing LLM serving frameworks (e.g., vLLM (17), SGLang (48)) leverage via continuous batching, paged attention, prefix caching, and priority-aware scheduling to boost throughput for model providers and reduce latency for users.

Figure 1 illustrates a common end-to-end deployment for serving DiTs. First, each user request is defined by three parameters: (i) spatial resolution—ranging from 240p to full HD (1080p), (ii) the number of denoising steps, or model feed-forwards, required for the desired fidelity, and (iii) video duration (2–60 s) ①. Together, these dimensions determine the total number of visual patches to be encoded for the transformer—which grows quadratically with resolution and linearly with frame count ②. DiT-Serve's central scheduler then tracks all active requests and dynamically schedules them onto data-parallel (DP) replicas of the diffusion transformer model ③. When an individual request's total patches exceeds the memory of a single replica (e.g. long, high-resolution videos), DiT-Serve invokes sequence parallelism, which slices long-context patches across DP instances to perform inference (23; 49).

Naturally, to efficiently serve video and image models, service providers must batch requests together to fully utilize their GPUs. However, batching heterogeneous DiT requests exposes two key inefficiencies, across space and time, in Fig. 2. First, temporal underutilization occurs when videos that require different numbers of denoising steps are batched together: GPUs processing the shorter requests finish early and then sit idle until the longest request completes. In Fig. 2a, requests A, B, and C require 2, 6, and 4 denoising steps respectively; due to heterogeneous steps, GPUs handling A and C idle for 4 and 2 units of time while waiting on B. Second, spatial underutilization happens when requests differ in resolutions or duration, resulting in patch sequences of varying lengths. Sequence-parallel schemes, like Ring Attention (23), pad every sequence to the maximum length, wasting compute and memory across GPUs. In Fig. 2b, A, B and C's patch sequence spans 8, 4 and 2 GPUS respectively, but all three are padded to 8 GPUs, leaving 50–75% of each batch's capacity unused.

We introduce DiT-Serve, an end-to-end serving system for diffusion transformers that efficiently batches heterogeneous video and image workloads to maximize GPU utilization. To eliminate temporal underutilization, DiT-Serve employs *step-level batching*, which preempts requests at each denoising step rather than running them to completion[1]. In Fig. 2a, shorter jobs (e.g., A and C) can yield GPU slots to pending tasks (e.g., D and E) while longer jobs (e.g., B) continue running. To mitigate spatial underutilization, we propose *Brick Attention*, which extends Ring Attention (23) by binpacking requests of different lengths into multiple independent rings of varying sizes, so that each request maps to its own ring. In Fig. 2b, Brick Attention would pack B and C onto separate rings totaling 6 separate GPUs, freeing remaining capacity for additional requests. Finally, DiT-Serve's scheduler employs a Shortest-Job-First (SJF) style policy: it ranks requests by their estimated compute and memory footprint—fewer denoising steps times patch sequence length—so that lightweight jobs complete quickly rather than being blocked by larger, long-running videos. Together, these techniques fully multiplex compute and memory across concurrent inferences, delivering cost-effective, high throughput performance.

---

[1]Analogous to continuous batching for LLM serving.

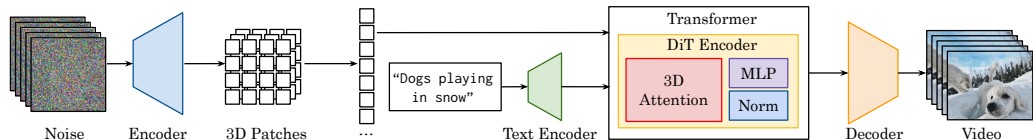

Figure 3: **Diffusion Transformer (DiT) Architecture.** Starting from a noisy latent tensor, an encoder extracts a 3D grid of spatio-temporal patches and flattens them into a sequence of visual embeddings. These tokens are concatenated with embeddings of the user's text prompt—obtained via a text encoder—and passed through a series of DiT encoder blocks that apply 3D self-attention across multiple denoising steps. Finally, a decoder upsamples the transformer outputs to reconstruct the denoised video.

We implement a prototype of DiT-Serve as a FastAPI server atop a production-ready, PyTorch-powered engine, and evaluate it on three state-of-the-art video diffusion models (Open-Sora (49), Mochi (40), CogVideoX (43)) under realistic, heterogeneous workloads. Our results show that DiT-Serve achieves up to 2-3× higher throughput and 3-4× lower latency compared to prior serving systems.

In summary, the primary contributions of this paper are:

- DiT-Serve is the first to integrate video and image generation as a service across user applications.
- We propose *step-level batching*, which preempts requests at each denoising step, and a novel sequence-parallel algorithm, *Brick Attention*, that binpacks requests of heterogeneous lengths.
- Our system is easily deployable, seamlessly integrates with FastAPI, supports many existing models, and demonstrates significant throughput and latency gains.

## 2 BACKGROUND & RELATED WORK

To detail relevant context for DiT-Serve, we provide a brief overview of Diffusion Transformers, state-of-the-art models, and existing serving systems.

### 2.1 DIFFUSION TRANSFORMERS

**Diffusion Models.** Diffusion models have emerged as a powerful class of generative models that achieve state-of-the-art performance across image, video, and 3D model generation. They assume a feedforward process that gradually applies Gaussian noise to real data $x_0 \sim q(x)$:

$$q(x_t \mid x_0) = \mathcal{N}\big(x_t;\ \sqrt{\alpha_t}\,x_0,\ (1 - \alpha_t)\mathbf{I}\big),$$

where $\alpha_t$ defines the standard deviation. Conversely, diffusion models are trained to learn the reverse the diffusion process, where a neural network ($\theta$) learns $p_\theta(x_{t-1}|x_t) = \mathcal{N}(\mu_\theta(x_t), \Sigma_\theta(x_t))$, iteratively applying $K$ denoising steps. Prior to diffusion transformers (DiTs), many models employed a U-net architecture (33; 34; 10). Current state-of-the-art video and image models integrate transformer encoder blocks (41), which applies full bidirectional, 3D attention across all patches (11; 49; 5; 29).

Figure 3 illustrates the general inference process for *text-to-video* diffusion models. Starting from a randomly sampled 3D Gaussian noise, an encoder extract a 3D grid of spatial–temporal patches, which is later flattened and concatenated with token embeddings produced by a text encoder. The combined sequence is passed through a transformer model, typically a transformer encoder (41), over $K$ passes, defined as denoising steps. Finally, the latent embedding is fed through a decoder, which upsamples and assembles the latent embedding back into a high-fidelity image or video.

### 2.2 VIDEO MODELS

Our engine supports three popular open-sourced diffusion transformers, which differ via their encoding scheme and transformer computation.

**Open-Sora (49)**. Open-Sora replaces the full 3D, patch attention with a two-stage Spatial-Temporal Diffusion Transformer (STDiT). First, a 3D variational autoencoder (VAE) (16) compresses a latent 3D embedding by 8× over spatial resolution and 4× over time. Second, Open-Sora applies two forms of attention: spatial attention, which attends over image resolution, and temporal attention,

which attends over video frames. Decoupling space and time for attention dramatically reduces the computation and memory requirement compared to full 3D attention.

**Mochi (40).** Mochi 1's AsymmDiT applies a causally-structured VAE (16) to compress a latent 3D embedding by 8× over spatial resolution and 6× over time For it's attention computation, AsymmDiT concatenates the spatial–temporal patch embeddings and text token embeddings into a single sequence and applies full bidirectional self-attention over the combined embeddings. Importantly, it allocates more attention heads to the visual patch tokens than to the textual tokens, prioritizing image information.

**CogVideoX (43).** CogVideoX 1.5 combines a 3D-causal VAE (8× spatial, 4× temporal) with causal masking in the latent encoder, employs an Expert Transformer that concatenates T5 text embeddings (32) and patch embeddings, and runs full 3D self-attention over the combined sequence. Uniquely, it applies separate adaptive LayerNorm parameters for text and video embeddings.

## 2.3 SERVING SYSTEMS

Much of DiT-Serve's innovations follow suite from popular LLM serving systems (i.e. vLLM (17), SGLang (48)). Recent innovations in LLM serving optimize memory management, kernel optimization, and scheduling (26; 39). To better manage memory, existing solutions, such as Orca and vLLM, integrate continuous batching and paging techniques to reduce KV-cache fragmentation (44; 17), introduce shared memory to cache prefixes across LLM requests (22; 48), and manage cache hierarchies between GPU, CPU, and disk (52; 38; 37). For improved kernels, other works improve the underlying CUDA kernels to accelerate attention (8), pipeline different operators (52), or implement different forms of parallelism (42; 19). Finally, LLM engines can leverage better scheduling, such as binpacking prefills and decodes together (1) and preempting LLM requests (42), to improve response times.

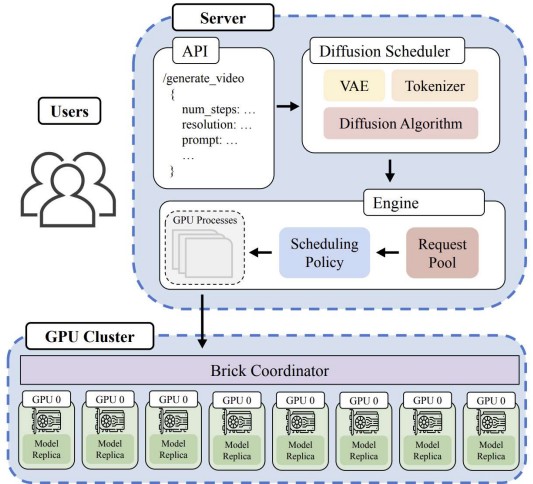

Figure 4: **DiT-Serve Architecture.** Execution flow begins with the FastAPI interface; each coroutine runs its own Diffusion algorithm, submitting requests to the engine when needed, which then distributes requests using the brick coordinator.

DiT-Serve core contributions align with innovations found in LLM scheduling. In particular, *step-level batching*, which preempts requests at step-level, is similar to continuous batching (44). DiT-Serve's *Brick Attention* binpacks requests onto GPUs, which better manages memory. Finally, the scheduler employs a preemptive scheduling policy, similar to prior LLM serving schedulers (42; 24). In the diffusion space, the closest work, xDiT (11), also addresses the inefficiencies of DiTs over multi-GPU clusters. xDit employs sequence and pipeline parallelism through PipeFusion and CFG parallelism. However, this work solely focuses optimizing a single request's latency, while DiT-Serve tackles optimization across requests, which vary via denoising steps, resolution, and time.

## 3 DIT-SERVE DESIGN

We present DiT-Serve's overall architecture and then explore its two key components: (1) a step-level request coordinator and (2) a distributed, multi-GPU attention algorithm for bin-packing requests with diverse context lengths.

## 3.1 OVERVIEW

DiT-Serve is a online serving engine that processes text-to-video Diffusion Transformers. DiT-Serve focuses on two primary objectives: (1) improving the end-to-end latency of each video generation request and (2) maximizing GPU utilization for providers.

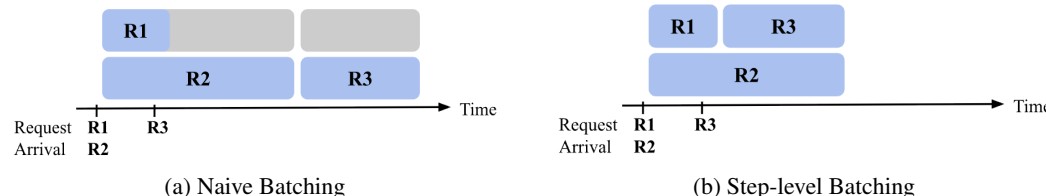

(a) Naive Batching     (b) Step-level Batching

Figure 5: **Batching Strategies.** (a) *Naive Batching*: Requests are grouped in fixed-size batches regardless of arrival time, leading to idle periods and underutilized GPU resources when shorter requests finish earlier. (b) *Step-level Batching*: With step-level batching, temporal GPU utilization is improved as request 3 can be batched together with request 2.

**Architecture.** Figure 4 illustrates DiT-Serve's overall architecture. Users interact with the system through Fast API calls, which sends requests to the Diffusion Scheduler that executes a defined scheduling policy[2]. The engine manages data parallel (DP) replicas of the DiT model and, similar to vLLM (17), runs a core engine loop that continually processes available requests provided by the scheduler. These requests are sent to dedicated GPU processes that use Brick Attention for processing variable context lengths across requests.

## 3.2 STEP-LEVEL BATCHING

DiT-Serve introduces *step-level batching*, a fine-grained scheduling strategy inspired by efficient memory management in large language model (LLM) serving (18; 44). Unlike existing diffusion serving systems—which treat each request as an indivisible job that must run to completion—our approach exploits the fact that each generation comprises of $K$ denoising steps. At every timestep, the scheduler selects the highest-priority requests, executes their current denoising step in a single GPU batch (Fig. 5b), and then returns completed requests to the pool. By continuously injecting new high-priority requests and evicting requests that have finished a step, step-level batching eliminates idle GPU cycles, maximizes GPU utilization, and yields substantially higher throughput compared to monolithic batching methods.

## 3.3 DISTRIBUTED ATTENTION

Due to long context length of long videos (1M-2M tokens), DiT-Serve employs sequence parallelism (SP) to chunk long sequences across GPUs, which prevents potential OOMs for GPUs and improves request latency. Existing SP implementations (i.e. Ring and Stripe Attention (23; 4)) do not consider the heterogeneity of requests and pad sequences to the longest context length. Hence, we introduce *Brick Attention*, a generalization of Ring Attention.

### 3.3.1 RING ATTENTION

Ring Attention (23) is an attention mechanism initially designed to overcome the limitations of training transformers over long contexts. Ring Attention distributes the input sequence across multiple devices in a ring-like topology. Each device processes a portion of the sequence, and the key-value pairs are incrementally passed along the ring, terminating when all key-value pairs have visited all GPUs. This allows for efficient computation of attention over a large sequence without materializing the entire attention matrix, thereby avoiding quadratic memory requirements and reducing model latency.

### 3.3.2 BRICK ATTENTION

We note that batching with Ring Attention assumes all requests have similar context lengths. For our system, requests vary based on resolution and duration; hence, context length varies significantly. In this scenario, Ring Attention would pad the shortest requests to the longest context, wasting GPU memory and computation (Fig. 2b).

---

[2]DiT-Serve supports a wide range of scheduling policies, such as First-Come-First-Serve (FCFS), Short Job First (SJF), and SRPTF (Shortest Remaining Processing Time First).

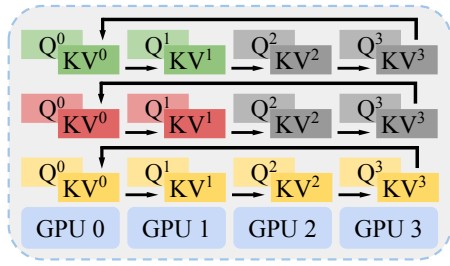

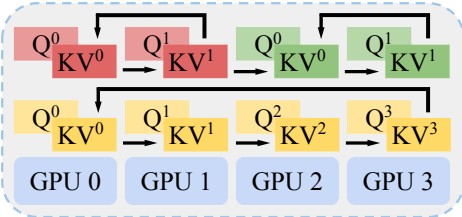

(a) Ring Attention                          (b) Brick Attention

Figure 6: **Distributed Attention.** (a) *Ring Attention*: Ring Attention falls short when considering the need to pad shorter requests together with longer ones; padding leads to wasted computation (gray blocks). This example contains three requests, one that takes four GPUs, and two that take two GPUs. (b) *Brick Attention*: Brick Attention supports batching of diverse context length requests by sending KVs to the correct devices for each ring. This example contains three requests, one that takes four GPUs, and two that take two GPUs.

*Brick Attention* extends Ring Attention by building multiple rings of different sizes—e.g., rings of 1, 2, 4, and 8 GPUs—over the same hardware, so that each GPU can serve several rings in parallel. We assign each request to its own ring whose size is proportional to its context length: on an eight-GPU cluster, the longest requests occupy the full eight-GPU ring, while shorter ones use smaller rings scaled accordingly. Finally, we apply a best-fit bin-packing algorithm across batch sizes and GPU resources to maximize utilization. By restricting ring sizes to powers of two, Brick Attention both simplifies the packing process and guarantees predictable performance.

We modify the KV communication logic so that each ring's attention computation does not interfere with each other. Attention calculation is still performed in one batch for efficiency, but cross-GPU KV communication for each batch is processed independently, allowing each device to send and receive KV to multiple different devices (Fig. 6b). This essentially creates, within each batch, multiple overlapping rings that each house the computation for a single request. Similar to Ring Attention, Brick Attention pipelines computation and KV communication. Due to multiple smaller rings and a sophisticated coordination logic, Brick Attention significantly improves resource utilization and scalability for heterogeneous requests.

### 3.4 SCHEDULING POLICY

Average request latency is highly sensitive to scheduling (see Fig. 10a). While classical queueing theory identifies **Shortest Job First** (SJF) as optimal, the notion of "shortest" must be reinterpreted for video-diffusion workloads on multi-GPU clusters. We therefore implement **Shortest Remaining Processing Time First** (SRPTF). For a request at diffusion step $t$, let $T_{\text{total}}$ denote total diffusion steps and $S(\text{req}_t)$ denote tokenised sequence length of the video at step $t$.

The remaining work for the request is defined as $L(\text{req}_t) = (T_{\text{total}} - t) S(\text{req}_t)$, or the product of the denoising steps still to run and the sequence length yet to process. Finally, requests are ordered by ascending $L(\text{req}_t)$; the request with the smallest remaining work executes first, mirroring SJF's theoretical optimality for job completion times.

## 4 EVALUATION

In this section, we evaluate DiT-Serve's performance across three leading video diffusion models and conduct ablations to analyze its behavior under varying workloads, its scalability with increasing GPU resources, and the effectiveness of its scheduling and batching strategies.

### 4.1 WORKLOADS

We simulate realistic workloads representative of practical usage scenarios encountered by video diffusion models. Specifically, synthetic requests are generated as follows:

---
**Algorithm 1** Brick Attention
---
**Require:** List of input tensors $X = \{X^{(1)}, X^{(2)}, \ldots, X^{(M)}\}$ with varying lengths
**Require:** Number of devices $D$
**Ensure:** Output tensors $\{Y^{(1)}, Y^{(2)}, \ldots, Y^{(M)}\}$
1: $X.sort\_descending()$
2: $batch \leftarrow 0, cur\_len \leftarrow 0$
3: $L_{\max} \leftarrow max(X[0].length)$
4: **for** each sequence $x$ in $X$ **do**
5:      $block\_size \leftarrow 2^{\left\lceil \log_2 \left( \frac{x}{L_{\max}} \right) \right\rceil}$
6:      Compute $Q, K, V$ with $x$
7:      $devices \leftarrow [cur\_len, \ldots, cur\_len + block\_size - 1]$
8:      $[Q_{batch,i}], [K_{batch,i}], [V_{batch,i}]_{i \in \text{devices}} \leftarrow split(Q, K, V)$
9:      $cur\_len \leftarrow cur\_len + block\_size$
10:      **if** $cur\_len = D$ **then**
11:          $cur\_len \leftarrow 0$
12:          $batch \leftarrow \text{batch} + 1$
13:      **end if**
14: **end for**
15: **for** each device $i = 1$ to $N$ **in parallel do**
16:      $KV \leftarrow [(K_{k,i}, V_{k,i})_k | k \in \{0, \ldots, \text{batch} - 1\}]$
17:      **for** each block $j = 1$ to $N$ **do**
18:          **for all** $k \in \{0, \ldots, \text{batch} - 1\}$ **do** $\text{send}(KV_k)$
19:          $A_j = \text{Attention}(Q_i, KV)$
20:          **for all** $k \in \{0, \ldots, \text{batch} - 1\}$ **do** $KV_k \leftarrow \text{recv}(k)$
21:      **end for**
22:      $Y_i \leftarrow \text{Combine outputs } A_{1-N}$
23: **end for**
24: Concatenate all $Y_i$ to form $Y$
---

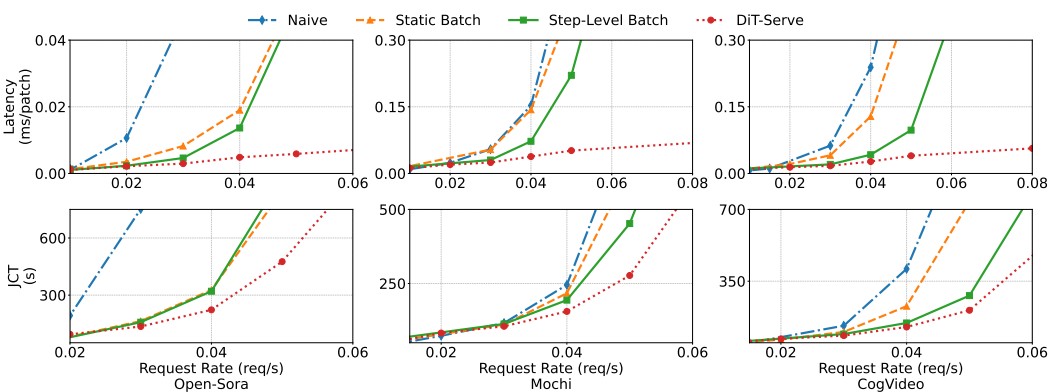

Figure 7: **Main Results.** Average latency and job completion time for different Video Diffusion models.

**Arrival Distribution.** Request's interarrival times follow an exponential distribution with varying rates ($\lambda$), simulating different levels of request frequency. Different arrival distributions are ablated in Fig. 10b.

**Request Specifications.** Each request randomly selects parameters to mimic realistic user scenarios:

- **Prompts:** Chosen from 300 textual prompts such as "A beautiful waterfall" or "A Chinese Lunar New Year celebration video with a Chinese dragon."
- **Resolution:** Uniformly sampled across [240p, 360p, 480p, 720p].
- **Sampling Steps:** Uniformly sampled across [20, 40, 80, 120] denoising steps.
- **Frame Count:** We use each model's default frame counts: 41 frames for CogVideoX (43), 31 frames for Mochi (40), and 48 frames for Open-Sora (49).

## 4.2 EXPERIMENT SETUP

Figure 8: **Tail Latencies.** 95th (P95) and 99th (P99) percentile latencies of different models.

**Models and Testbed.** We evaluate three state-of-the-art video diffusion models: CogVideoX (43), Mochi (40), and Open-Sora (49), configured to utilize 1, 2, 4, and 8 GPUs, respectively. Experiments are conducted on a Google Cloud Platform Compute Engine `a2-ultragpu-8g` instance, featuring eight interconnected A100-SXM4-80GB GPUs via NVLink, 1360 GB host memory, PCIe-4.0×16 connectivity, and 2 TB storage capacity.

**Metrics.** Our primary metric measures *normalized latency*, measured as requests' end-to-end response time divided by the number of patch computations, defined as the product of denoising steps times patches per step. An efficient, high-throughput system should minimize normalized latency, even under high request rates. For the rest of evaluation, this metric simply referred to as *latency*.

**Baselines.** To rigorously assess our system, we benchmark against four baselines, each constrained by identical maximum batch sizes of five for fairness:

- **Naive (Naive + Ring Attention + FCFS).** Naively processes requests individually without batching and assigns requests GPUs based on computational demand determined by resolution and frame count. Tasks follow a First-Come-First-Served (FCFS) approach, leading to head-of-line (HoL) blocking. It employs Ring Attention (23) to distribute computation across GPU clusters.
- **Static Batch (Naive Batching + Ring Attention + FCFS):** Groups similar requests (matching resolution and frame count) up to a max batch of five to alleviate head-of-line (HoL) blocking. Engine schedules requests based on FCFS order and employs Ring Attention (23) to schedule requests across GPUs.
- **Step-level FIFO Batch (Step-level Batching + Brick Attention + FCFS):** Employs step-level batching method, dynamically integrating incoming requests and promptly removing completed ones to enhance responsiveness. It leverages Brick Attention for improved handling of diverse requests across GPU clusters and maintains FIFO scheduling for fairness. Maximum batch size when binpacked with Brick Attention is five.
- **DiT-Serve (Step-level Batching + Brick Attention + SRPTF):** Enhances step-level batching with Brick Attention and adopts Shortest Remaining Processing Time First (SRPTF) scheduling. Maximum batch size when binpacked with Brick Attention is five. SRPTF prioritizes requests with the lowest remaining computational load, significantly reducing wait times and enhancing overall throughput and responsiveness.

## 4.3 END-TO-END PERFORMANCE

Figure 7 compares normalized latency and job completion times across scheduling and batching strategies for Open-Sora (49), Mochi (40), and CogVideoX (43). Our approach—Step Level Batching integrated with Brick Attention and SRPTF scheduling—significantly outperforms other baselines, consistently delivering the lowest latency and shortest completion times, particularly under moderate to high workloads. In contrast, Naive Scheduling and Static Batching exhibit notably higher latencies due to inefficient resource use and head-of-line blocking. For Open-Sora, our method reduces latency by approximately 3-4× compared to naive methods under moderate to heavy workloads. Similar improvements are noted with Mochi and CogVideoX, confirming the general efficacy of our method across diverse video diffusion models.

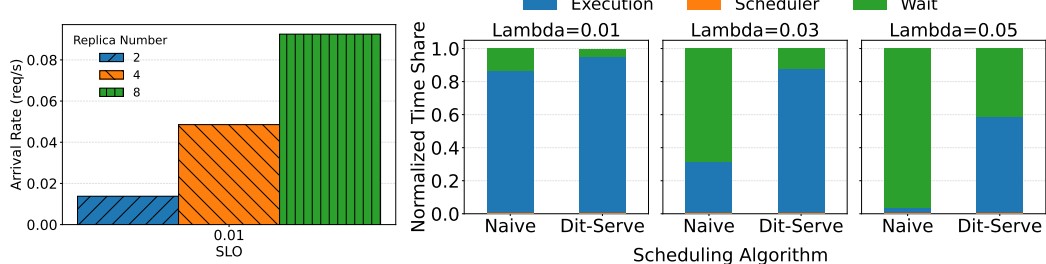

(a) **Scalability Experiments.**      (b) **Breakdown of Inference Overheads.**

Figure 9: **Scalability and Inference Overheads.** (a) Given same SLO (ms/patch), DiT-Serve's max arrival rate (req/s) scales linearly w.r.t number of GPUs. (b) This example from Mochi 1 Preview shows that DiT-Serve significantly reduces wait time and introduces trivial scheduler overheads. Scheduling time is negligible compared to execution and wait times.

**Tail Latency.** Figure 8 provides a detailed analysis of tail latencies (95th and 99th percentiles, P95/99) across all models. Step Level Batching with Brick Attention significantly reduces tail latency relative to Naive Scheduling and Static Batching, especially under intensive workloads. For instance, Open-Sora and CogVideoX demonstrate substantial decreases in P95 and P99 latencies, highlighting the robustness and effectiveness of our proposed system under demanding conditions. Similar trends are observed with Mochi, further validating our method's reliability and overall efficiency.

## 4.4    SCALABILITY.

We evaluate how well DiT-Serve scales with increasing GPU resources by measuring the maximum sustained request arrival rate that meets a fixed service-level objective (SLO). In this experiment, the SLO is defined as end-to-end latency of 0.01 ms per token on average. We vary the number of available GPUs from 2 to 8 and record the highest arrival rate (in req/s) at which the system can operate without violating the SLO. As shown in Figure **??**, DiT-Serve demonstrates near-linear scalability: doubling the number of GPUs consistently leads to a proportional increase in throughput. With 2, 4, and 8 GPUs, the system achieves maximum arrival rates of 0.02, 0.05, and 0.10 req/s respectively. These results highlight DiT-Serve's ability to efficiently utilize additional hardware and meet service requirements under growing workloads.

## 4.5    TIMING BREAKDOWN

Figure 9b breaks down the time that video generation requests spend in the serving layer for both DiT-Serve and the Naive baseline with head-of-line blocking. Overall, DiT-Serve consistently incurs significantly lower waiting time, primarily attributed to its Step Level batching strategy and the SRPTF scheduling policy. Although SRPTF introduces preemption overhead, the incurred scheduling cost is minimal and virtually imperceptible. In contrast, the Naive baseline suffers from substantial waiting time due to a FCFS scheudling policy and substantial lack of batching, which leads to severe head-of-line (HoL) blocking. Notably, as request arrival rate increases, the waiting time under the Naive policy grows sharply, while DiT-Serve maintains significantly better scalability, with only modest increases in waiting time. This highlights the benefits of fine-grained scheduling and coordinated attention mechanisms for reducing wait times in video serving systems.

## 5    CONCLUSION

We present DiT-Serve, a distributed Diffusion Transformer serving system designed to efficiently handle diverse, online text-to-video generation requests. By addressing both spatial and temporal inefficiencies, DiT-Serve leverages two core innovations to improve GPU utilization: step-level batching and Brick Attention. Step-level batching improves throughput by preempting requests at each denoising step, while Brick Attention enables efficient, batched sequence parallelism. Our experiments across three state-of-the-art video diffusion models demonstrate that DiT-Serve significantly reduces latency by 3-4x and increases throughput by 2-3x compared to prior systems.

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

# A  APPENDIX

## A.1  THE USE OF LARGE LANGUAGE MODELS

We used large language models (LLMs) exclusively for editorial assistance to improve clarity, grammar, and concision. Specifically, LLM support was limited to refining wording and reducing redundancy, reorganizing sentences for readability and flow, standardizing terminology and style, and minor copy-editing (spelling, punctuation, formatting). LLMs were not used for research ideation, literature review, methodological design, data analysis, result interpretation, or drafting novel scientific content. All scientific claims, framing, methods, results, and conclusions were conceived, written, and verified by the authors. The authors take full responsibility for the paper's content, including text refined with LLM assistance.

## A.2  IMPLEMENTATION

DiT-Serve is an end-to-end Diffusion Transformer serving system totaling 10k lines of Python code.

**Backend.** DiT-Serve's API is exposed using FastAPI. Each FastAPI coroutine submits model requests to the engine when needed. Any mathematical calculations needed for the Diffusion Scheduler are performed on the same coroutine, on GPU 0. These calculations have a negligible impact on performance, and are not optimized through batching across requests or multi-GPU inference.

**Engine.** The engine's model running loop runs as a coroutine on the same `asyncio` event loop that the FastAPI server runs on. The engine also initializes GPU communication protocols for Brick Attention at the beginning of every model call. Currently, DiT-Serve supports serving one model at a time and does not route across different models.

**System Initialization.** Upon server startup, the model weights are replicated onto each GPU. New processes are spawned to handle model execution for each GPU, and communication between the server and the processes is done through PyTorch Multiprocessing queues.

**Brick Attention.** The implementation of Brick Attention is based on an open-source PyTorch implementation of Ring Attention (53). It uses PyTorch Multiprocessing (MPI) for GPU communication, which leverages NVLink for fast inter-GPU communication for KV blocks. The interface of Brick Attention closely mirrors that of the Ring Attention implementation, which itself mirrors that of Flash Attention. Since Diffusion Transformers employ bidirectional attention, load balancing variants of Ring Attention like Zig-zag and Stripe Attention (4) were not implemented for Brick Attention.

**VAE Parallelism** Our engine incorporates Variational Autoencoder (VAE) parallelism (16; 49), inspired by techniques presented in recent Diffusion Transformer inference engines (11). This technique divides the latent feature maps into multiple patches, distributing them across GPUs to parallelize decoding computations, alleviating memory bottlenecks.

## A.3  COMPARISON TO OPTIMAL SCHEDULING

Besides SRPTF, we evaluate the impact of different scheduling policies within DiT-Serve on normalized latency and job completion time under varying request rates. We include the following policies:

- **FCFS** (First Come First Serve): Jobs that arrive first run first.
- **SJF** (Shortest Job First): Jobs with the lowest total number of denoising steps run first.
- **SRTF** (Shortest Remaining Time First): Jobs with the fewest remaining denoising steps run first.
- **SRPTF** (Shortest Remaining Process Time First): Sorted by remaining steps multiplied by context length.

Our experiments in Fig. 10a indicate that SRPTF consistently outperforms all other policies, achieving the lowest latency and shortest completion times. In contrast, FIFO performs worst due to significant head-of-line (HoL) blocking. SJF and SRTF provide intermediate performance, reflecting improvements from prioritizing shorter tasks dynamically. These results underline the advantage of clairvoyance-based scheduling, highlighting DiT-Serve's optimality.

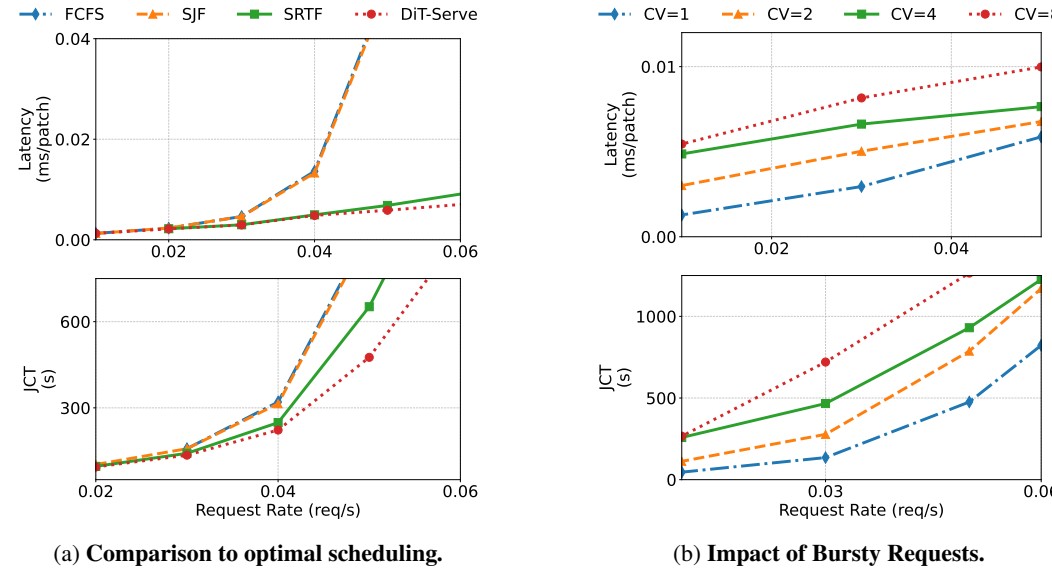

(a) **Comparison to optimal scheduling.**       (b) **Impact of Bursty Requests.**

Figure 10: **Scheduling Ablations.** (a) DiT-Serve with SRPTF outperforms other scheduling policies. (b) We simulate burstiness in request arrivals using a Gamma Distribution with varying coefficient of variation (CV). DiT-Serve maintains robust performance across varying levels of request burstiness.

## A.4 ARRIVAL DISTRIBUTION/WORKLOAD

We ablate the effect of burstiness by varying the coefficient of variation (CV) of request inter-arrival times, drawing requests from a Gamma Distribution. This models a range from relatively uniform (CV = 1) to highly bursty (CV = 8) arrival processes to capture variability observed in production-scale inference workloads. Figure 10b shows the effect of varying burstiness on normalized latency and average job completion time (JCT). We observe that across all CVs, both latency and JCT remain relatively low when arrival rate is low. As the rate increases, however, high-CV workloads generate sudden request bursts that lengthen queues and degrade performance.

## A.5 OFFLINE INFERENCE.

In many real-world scenarios, video diffusion workloads may not always arrive online; but instead in large, queued batches. To evaluate DiT-Serve's performance under such "offline" scenarios, we simulate the processing of a fixed number of video generation requests, all arriving simultaneously at time zero. The key metric of interest is makespan, defined as the time between the start of the first request's processing and the completion of the last request. Figure **??** presents the makespan across four baselines under offline batch settings. DiT-Serve with Brick Attention

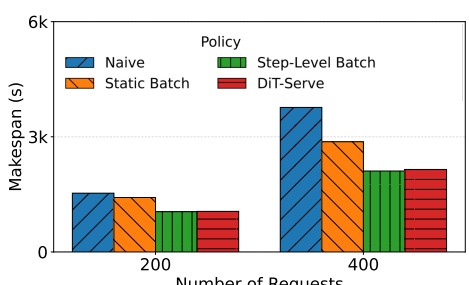

Figure 11: **Offline batch inference.** DiT-Serve decreases the makespan to process a batch.

consistently achieves the lowest start-to-finish times, reducing makespan by 25–65% compared to other baselines. Together, these results highlight DiT-Serve's efficiency in offline, high-throughput environments.