# OpenReview forum: "DiT-Serve: An Efficient Serving Engine for Diffusion Transformers"
_ICLR.cc/2026/Conference — Submitted to ICLR 2026_

### Official Review · Reviewer_Gfvy · 2025-10-29

**Soundness:** 3
**Presentation:** 3
**Contribution:** 2
**Rating:** 4
**Confidence:** 5

**Summary:**

The paper studies the problem of servicing systems for diffusion transformers. The paper proposes DiT-Serve, a system with two major components: 1) DiT-Serve batches diffusion jobs at the step level and 2) DiT-Serve utilizes a brick-attention mechanism, which is based on the ring attention, to perform distributed attention computation. The job scheduling of DiT-Serve is shortest-job first, and it identifies the shortest job based on its remaining computational costs. Empirically, DiT-Serve shows lower latency compared with baseline methods.

**Strengths:**

1. The paper aims to solve a significant and practical problem of serving large diffusion transformer models.

2. The paper shows empirical improvements compared to baselines.

**Weaknesses:**

1. The novelty of the paper is not quite significant. Step-based batching is quite a natural solution in the context of diffusion model serving, and the brick attention mechanism essentially runs multiple instances of the ring attention.

2. The system utilizes a shortest job first strategy for scheduling, and it has no mechanism for the deadline of service, so a long job may have a long wait before its completion.

3. Line 290-291 is very briefly described, and I think more details about the optimization objective should be provided. Also, given the variations in video resolution and video length, the cost of each sequence may vary quite significantly. However, the current design supports only block sizes of 1, 2, 4, and 8, which may still require significant padding and thus result in wasted computation.

4. It would also be valuable to add the GPU FLOPs utilization metric to the experiments.

Minor:
1. Line 091: " GPUS"
2. Line 332: x should be the length of x
3. Line 460: "Figure ??"
4. Line 746: "Figure ??"

**Questions:**

Please address the weakness part.

---

### Official Review · Reviewer_yzp7 · 2025-10-29

**Soundness:** 3
**Presentation:** 2
**Contribution:** 3
**Rating:** 4
**Confidence:** 2

**Summary:**

This paper introduces **DiT-Serve**, an efficient serving engine for Diffusion Transformers (DiTs) designed to handle heterogeneous requests varying in resolution, video length, and denoising steps. The authors identify two key inefficiencies in current systems: **spatial underutilization** due to padding and **temporal underutilization** from batching jobs with different denoising step counts. To address these, DiT-Serve proposes two main contributions: **step-level batching**, which preempts and swaps requests at each denoising step , and **Brick Attention**, a novel attention algorithm that bin-packs requests of different context lengths onto multiple GPUs to reduce padding. Evaluations on several state-of-the-art models show that DiT-Serve achieves 2-3x higher throughput and 3-4x lower latency compared to prior systems.

**Strengths:**

- **Timely and Practical Perspective**: The paper tackles the important and practical problem of efficiently serving diffusion models. Focusing on optimizing for heterogeneous request workloads, which mirror real-world use, is a valuable contribution to the community as these models move to production.

- **Clear Illustrations**: The figures are a significant strength. Figures 2, 5, and 6, in particular, provide clear and intuitive visualizations of the core problems (spatial/temporal underutilization) and the proposed solutions (Brick Attention, step-level batching), which greatly aids in understanding the paper's contributions.

- **Realistic Workload Simulation**: The evaluation methodology is strong. The use of an online serving simulation where request inter-arrival times follow an exponential distribution (and ablations with Gamma distributions ) provides a compelling and realistic proxy for a real-time service environment.

**Weaknesses:**

1. Formatting and Presentation Issues:

* **PDF Formatting**: The submitted PDF appears to have formatting issues. The standard "Under review as a conference paper at ICLR 2026" header and page numbers seem to be missing from the document's headers and footers.

* **Broken References**: There appear to be broken cross-references in the text, which show up as "??". I noted this in Line 460 and 746, referring to Figure ??. Please correct these.

* **Abstract Mismatch**: I observed a minor mismatch between the abstract provided on the OpenReview submission page and the abstract within the PDF itself. Please ensure these are consistent.

2. Comparison to Related Work (xDiT):

The authors identify xDiT as the "closest work" in the diffusion space. The paper argues it is not a direct baseline because it "solely focuses optimizing a single request's latency", whereas DiT-Serve focuses on multi-request optimization.
This distinction is fine, but a more thorough investigation seems warranted. A "Naive" (FCFS) scheduler running requests on top of xDiT's single-request optimizations might serve as a much stronger baseline than the "Naive + Ring Attention"  baseline used. Could the authors elaborate on why xDiT was not included as a baseline, or ideally, provide a comparison?

3. Reproducibility Concerns:

This paper describes a heavy distributed system (10k lines of Python code). For a systems paper of this nature, reproducibility is paramount to verify the claims and allow the community to build on this work.
The paper does not include a statement on whether the code will be made open-source. Given the complexity of implementing custom attention mechanisms (Brick Attention) and schedulers, the results are difficult to verify without access to the code. What are the authors' plans for a code release?

**Questions:**

See weakness.

---

### Official Review · Reviewer_mV7j · 2025-11-01

**Soundness:** 3
**Presentation:** 1
**Contribution:** 2
**Rating:** 2
**Confidence:** 3

**Summary:**

This paper introduces DiT-Serve, an efficient serving engine designed specifically for diffusion transformers (DiTs).
The authors propose step-level batching to handle heterogeneous workloads effectively and introduce Brick Attention, a distributed attention algorithm designed to reduce padding overhead.
These innovations improve GPU utilization and overall performance for serving image and video diffusion models.

**Strengths:**

- The engineering of the proposed system appears quite robust, especially the ideas around GPU batching and the introduction of Brick Attention for reducing padding.

**Weaknesses:**

- Several presentation issues hinder clarity: a typo on line 374, incorrect references in Section 4.4, and mistakes in Figure 3 (the noise shouldn't go through an encoder, and encoder-decoder diagrams seem reversed).
- No quantitative data tables are provided; the evaluations are purely visual via charts, making detailed numerical analysis difficult.
- The method presented lacks novelty; it feels much more like practical engineering than original research suitable for ICLR.

**Questions:**

- Do the authors plan to open-source their implementation?

---

### Official Review · Reviewer_wn71 · 2025-11-01

**Soundness:** 3
**Presentation:** 3
**Contribution:** 2
**Rating:** 4
**Confidence:** 3

**Summary:**

This paper proposes an efficient serving engine for diffusion transformers. The key challenge here is that various requests may have various number of tokens and denoising steps. Previous methods typically pad to the longest one, which results in poor utilization of GPU resources. This paper proposes 1) step-level batching, where requests are scheduled in a per-step basis, and 2) Brick Attention, which allocates a dynamic number of GPUs depending on each request. Experiments demonstrate that the proposed system yields higher throughput and lower latency compared to prior systems.

**Strengths:**

1. The topic of this paper is interesting, practical, and meaningful.
2. It is interesting to solve the problem in a best-fit bin-packing algorithm.
3. The writing is generally clear.

**Weaknesses:**

1. It appears somewhat naive to use denoising step as a unit for the scheduler. It would introduce a lot of offload efforts for transmitting data between GPU and CPU, especially when the models are also dynamic, which also involves model offload.
2. It may not be optimal to allocate an integral number of GPUs to each request. In many cases, the optimal solution can be allocate $n+0.5$ GPUs to request A and $m+0.5$ GPUs to request B.
3. The paper may be finished in a rush. There are a lot of typos. For examples, in Line -10 of the abstract, "which" should be "where". In Fig. 4, all the 8 GPUs are "GPU 0". And there is an undefined reference in Line 746 of the last page.

**Questions:**

I would like to further discuss with the authors about the first two points listed in the Weaknesses part, i.e., the cost of CPU offload, the potential solution when there are various requests use various models, the potential optimality of various requests sharing a GPU. I would happy to increase the score if these questions are well discussed.

---

### Meta-Review · Area_Chair_bhY9 · 2025-12-03

**Summary:**

This paper introduces DiT-Serve, a serving engine for Diffusion Transformers designed to improve efficiency for heterogeneous requests varying in resolution and denoising steps. The core proposals are step-level batching to reduce GPU idle time and a Brick Attention algorithm to minimize padding overhead. The evaluation reports significant gains in throughput and latency. The reviewers generally find the problem practical and the engineering solid. However, significant concerns are raised across all reviews. The primary issues are a perceived lack of novelty, with the solutions viewed as straightforward adaptations of existing concepts like step-wise scheduling and ring attention, making the contribution feel more like engineering than foundational research suitable for ICLR. Several presentation and formatting issues are noted, including typos, broken references, and missing quantitative data tables. Key technical concerns include the potential overhead of frequent CPU-GPU data transfers for step-level batching, the optimality of allocating whole GPUs versus sharing them fractionally, and the absence of a strong comparison to the related xDiT system. The authors' rebuttal attempts to clarify misunderstandings, such as the data transfer mechanism not involving full model offload, and commits to fixing presentation errors. However, the rebuttal does not fundamentally alter the perception of limited novelty or fully address concerns about the baseline comparison and the foundational research contribution. Two reviewers gave a borderline score of 4, one gave a clear reject (2), and another gave a 4. The consensus leans toward rejection due to insufficient novelty for the conference.

**Reviewer Concerns:**

The authors addressed some minor presentation issues in the rebuttal. However, the major concerns regarding the core novelty of the work, its nature as an engineering optimization rather than a research breakthrough, and the lack of a compelling comparison to xDiT remain outstanding.

**Reviewer Scores:**

Given the discussion likely reinforced concerns about novelty rather than alleviating them, it is probable that the reviewers who initially gave a score of 4 would maintain or slightly lower their score. The reviewer who gave a 2 would see no reason to change their score.

---

### Decision · Program_Chairs · 2026-01-26

Reject